# 3D Printing of a Self-Healing Thermoplastic Polyurethane through FDM: From Polymer Slab to Mechanical Assessment

**DOI:** 10.3390/polym13020305

**Published:** 2021-01-19

**Authors:** Linda Ritzen, Vincenzo Montano, Santiago J. Garcia

**Affiliations:** Novel Aerospace Materials Group, Faculty of Aerospace Engineering, Delft University of Technology, Kluyverweg 1, 2629 HS Delft, The Netherlands; L.Ritzen@tudelft.nl (L.R.); V.Montano@tudelft.nl (V.M.)

**Keywords:** self-healing, polyurethane, 3D printing, cut test

## Abstract

The use of self-healing (SH) polymers to make 3D-printed polymeric parts offers the potential to increase the quality of 3D-printed parts and to increase their durability and damage tolerance due to their (on-demand) dynamic nature. Nevertheless, 3D-printing of such dynamic polymers is not a straightforward process due to their polymer architecture and rheological complexity and the limited quantities produced at lab-scale. This limits the exploration of the full potential of self-healing polymers. In this paper, we present the complete process for fused deposition modelling of a room temperature self-healing polyurethane. Starting from the synthesis and polymer slab manufacturing, we processed the polymer into a continuous filament and 3D printed parts. For the characterization of the 3D printed parts, we used a compression cut test, which proved useful when limited amount of material is available. The test was able to quasi-quantitatively assess both bulk and 3D printed samples and their self-healing behavior. The mechanical and healing behavior of the 3D printed self-healing polyurethane was highly similar to that of the bulk SH polymer. This indicates that the self-healing property of the polymer was retained even after multiple processing steps and printing. Compared to a commercial 3D-printing thermoplastic polyurethane, the self-healing polymer displayed a smaller mechanical dependency on the printing conditions with the added value of healing cuts at room temperature.

## 1. Introduction

Additive manufacturing (AM), or three-dimensional (3D) printing, of smart polymers is a rapidly growing field [1,2,3]. The different AM techniques offer in principle the possibility to manufacture, often expensive, smart materials in a versatile, minimum-waste manner. Autogenous or intrinsic self-healing polymers are a type of man-made polymers that aim to extend the service life of products made thereof through the autonomous or on-demand repair of damages such as cracks or scratches. Through careful design of the polymer architecture, reversible or dynamic bonds can be implemented in the polymer network in a search for a balance between healing and sufficiently good mechanical properties [4]. These dynamic bonds can be intermolecular reversible covalent bonds, such as Diels–Alder cycloaddition and polysulphide reactions, or supramolecular, such as hydrogen bonding or ionic clustering. Although the recovery of mechanical and other functional properties such as barrier or electrical conductivity of self-healing polymers is the subject of significant attention, the translation of such concepts to 3D printed parts has received very limited attention, mostly focused on the printability of a couple of polymers with limited attention to the study of the mechanical properties or healing potential of the printed parts [3]. 

The interlayer adhesion between deposited filaments using regular thermoplastic polymers relies mostly on tangling of the polymer chains. This tangling is often limited by the high viscosity of the polymers used and because the layer previously deposited has already cooled down upon deposition of the next one. In healing polymers, the dynamic bonds that facilitate the healing process may promote interlayer adhesion during printing, thereby resulting in parts with reduced anisotropy [5,6,7]. 

Fused deposition modelling (FDM) and liquid deposition modelling (LDM) are the most common 3D-printing methods of polymers. Of these, fused deposition modelling (FDM), or fused filament fabrication (FFF), is the most popular method on both the amateur and professional level as it allows continuous printing [8]. FDM machines are melt-extrusion based and use a polymer continuous filament as the input material [9]. The filament is pushed by gears into the feeder system through a filament guide towards the heater block. This block is heated to a set temperature such that the material can be molten and extruded through the nozzle. The 3D printed material is then created layer by layer on a print bed that may also be heated. Liquid deposition modelling (LDM), on the other hand, uses a (heated) syringe instead of a (heated) printing head to locally melt the filament. This process does not require a filament but has other drawbacks such as the reduced volume of the syringe, which limits continuous printing, and the long residence time of the material at elevated temperatures, which may affect the polymer integrity [9]. The main drawback FDM compared to LDM is that it requires the availability of a polymer that can be produced at relatively large amounts and available as a continuous filament bobbin. Unfortunately, this is uncommon for lab-scale proof-of-concept polymers such as those subjects of study in self-healing and smart polymer research. As a consequence, extrusion-based 3D printing of intrinsic self-healing polymers has mostly focused on LDM printing of hydrogels [10] and temperature-reactive healable polymers based on reversible covalent bonds such as Diels–Alder [5,6,7] or ionic interactions [11], requiring high (≥120 °C) healing temperatures, and, to a minor extent, elastomers based on hydrogen bonding still healing at temperatures above 80 °C [12]. Little to no attention has been put on the printing of self-healing thermoplastic polymers able to heal at moderate or near room-temperature relying on non-covalent interactions for the healing process such as disulfide bonds [13,14], hydrogen bonding [15,16], or combinations thereof such as hydrogen bonds and aromatic interactions [17,18]. These polymers show high melt adhesion and viscosity which make their processability difficult, but their well-studied healing ability and underlying mechanisms and the relatively large amounts at which they can be produced position them as good candidates to explore FDM of self-healing polymers. 

The quality of the 3D printed parts is generally assessed optically and, to a lesser extent, mechanically. The ASTM D638 tensile test [19] is probably the most commonly used mechanical test to study the tensile behaviour and the anisotropy of 3D printed parts [5,6,7]. However, smart polymers and lab-scale experimental polymers are often manufactured in small quantities, thereby not complying with the necessary large material amounts used in common mechanical testing. Mechanical tests able to give (pseudo)quantitative information of small 3D-printed parts would be beneficial for the full exploration of experimental polymers. A mechanical test requiring small amounts of material is the compression cut test developed for elastomers [20]. In this test, an elastomer is compressed between a plateau and a blade which cuts the sample until failure. The compression-cut test gives information about the compressive behaviour and failure of relatively small samples and has a low dependency on the sample dimensions compared to other mechanical testing methods [20]. This test is largely used in the food industry to study soft tissues [21], but it has not yet been implemented in the study of self-healing materials or 3D printed polymeric parts to the best of our knowledge.

Thermoplastic polyurethanes (TPU) have attracted significant attention in the field of self-healing polymers due to the interesting combination between easily accessible healing conditions (room-temperature healing) and the decent mechanical properties they offer. Although healing can be achieved through different reversible chemistries [22], in PUs, interfacial healing is mostly efficiently achieved through hydrogen bonding among urethane units at the broken interface. Despite highly efficient healing being able to be achieved using this strategy, the high reversible bond density can also lead to the formation of micro-crystals and to the suppression of macromolecular dynamics and therefore healing. To find autonomous healing polyurethanes with interesting mechanical properties and a good balance between healing dynamics and network stability without crystallinity, Yanagisawa et al. [23] inserted asymmetrical and highly dynamic thiourea bonds in the main chain, while Montano et al. regulated main chain dynamics via lateral fully aliphatic dangling chains [24,25].

In this work, we report a protocol to 3D-print and mechanically test a previously reported [25]) low-temperature self-healing thermoplastic polyurethane (SH-TPU) using fused deposition modelling. The SH-TPU was synthesized as a polymer slab, processed into a filament using a commercial filament maker and successfully printed with a modified commercial 3D printer. In order to characterise the mechanical properties of the 3D printed self-healing polyurethane, a compression cut-test based on a previously reported test for elastomers [20] was used. The mechanical properties of the 3D printed parts were compared to the bulk polymer and a comparable non-healable commercial polymer used for 3D printing. 

## 2. Materials and Methods 

### 2.1. Materials

The self-healing thermoplastic polyurethane (SH-TPU) used in this work was synthesized using a long diol (CroHeal™2000) provided by Croda Nederland B.V. (Croda Nederland B.V., Gouda, The Netherlands), 4,4′-Methylenebis(phenyl isocyanate) (MDI), 98%, from Sigma-Aldrich (Sigma-Aldrich, Zwijndrecht, The Netherlands) as diisocyanate, and 2-Ethyl-1,3-Hexanediol (EHD), 97%, mixture of isomers, from Sigma-Aldrich (Sigma-Aldrich, Zwijndrecht, The Netherlands) as the chain extender. The chemical structure of the components can be seen in Figure 1. 

The first step of the synthesis process was to warm up the oligomer CroHeal™2000 at 75 °C for 2 h to reduce its viscosity. Then, 90 g of CroHeal™2000 was transferred into a polypropylene cup and placed back into the oven at 60 °C. The weight of the other components was calculated based on the weight of CroHeal™2000 (Molar ratio CroHeal™2000:EHD:MDI = 1:0.6:1.7). When the CroHeal™2000 had been in the oven for 40 min, the MDI was weighed into a glass jar, adding 1 g to compensate for residue after pouring. EHD was then added to CroHeal™2000. The CroHeal/EHD diol mixture and the MDI were subsequently placed in an oven at 55 °C for roughly 45 min, or until the MDI had fully liquefied. Then, both containers were removed from the oven and the MDI was added to the diol mixture on a scale. The diol-isocyanate was then placed in a high speed mixer (SpeedMixerTM DAC 400.2 VAC-P (Speedmixer, Buckinghamshire, UK)), degassed to 40 mbar for 2 min and then mixed at 1800 rpm for 45 s. Once removed from the speed mixer, the mixture was poured into a petri-dish lined with nylon foil and placed into an oven at 65 °C for 18 h. The SH-TPU polymer synthesis, characterization and healing behaviour can be found in more detail in [25], where the segmented polyurethane appears referenced as MDI-p. A commercial thermoplastic PU (semi-transparent Ninjaflex (Ninjatek, Manheim, PA, USA)) filament with a diameter of 1.75 mm was used as a regular TPU reference for having comparable, yet superior, mechanical properties.

### 2.2. Characterisation Techniques 

A Spectrum 100 FTIR spectrometer from PerkinElmer (PerkinElmer, Waltham, MA, USA) was used to conduct Fourier transform infrared spectroscopy (FT-IR) measurements in representative areas. Spectra were conducted from 4000 cm^−1^ to 500 cm^−1^ using the average of 8 scans.

A TGA 4000 from PerkinElmer (PerkinElmer, Waltham, Massachusetts, USA) purged under nitrogen atmosphere was used for thermogravimetric analysis (TGA). The TGA was run from 30 °C to 600 °C at 5 °C/min. 

Differential scanning calorimetry (DSC) measurements were conducted with a DSC 250 by TA Instruments (TA Instruments, New Castle, UK), using an empty pan as a reference. DSC measurements were performed under dry nitrogen atmosphere at 5 °C/min heating and cooling rates over the range from −30 °C to 250 °C.

Small amplitude oscillatory shear analyses were conducted on a Haake Mars III (ThermoFisher Scientific, Waltham, MA, USA) rheometer with flat plates with a diameter of 8 mm. Three types of measurements were conducted. Amplitude analyses were performed in stress control mode from 10 to 1000 N/m^2^ at a frequency of 1 Hz to assess the linear viscoelastic regime of the studied polymer. Temperature sweep analyses were performed heating from 30 °C to 250 °C followed by cooling back to 30 °C. Scans were performed at 1 °C/min at a frequency of 1 Hz and at a constant shear strain of 0.1%. Shear rate analyses were performed at different temperatures, from 195 °C to 240 °C, with 5 °C increments. The shear rate was varied from 1 × 10^−3^ 1/s to 1 × 10^3^ 1/s logarithmically. The shear rate analysis was repeated for Ninjaflex, with the temperature ranging from 205 °C to 235 °C in 5 °C increments.

Optical microscopic pictures were taken using a Keyence VHX-2000 (Keyence, Osaka, Japan).

### 2.3. Filament Making

The filament of the SH-TPU was prepared using a 3DEVO Next 1.0—ADVANCED (3DEVO B.V., Utrecht, The Netherlands) filament maker. The machine came equipped with a single nitride extruder screw and 4 heating zones that can be set individually (T1 to T4, where T1 is closest to the exit of the extruder). The screw rotation speed could be adjusted in order to control the volumetric flow rate in the extruder. As the screw rotation speed also influences the material viscous response, the selection of the screw rotation speed had to be adjusted to the temperatures used in the extruder in order to ensure a sufficient material feed. Two fans with adjustable fan speed and orientation helped the rapid cooling of the filament at the exit of the extruder section. By properly controlling the temperature of the polymer right at the exit of the extruder section, the filament diameter could be controlled more easily, resulting in smooth filaments with more or less constant diameter. The diameter of the filament is further controlled by a puller system, the speed of which was automatically adjusted based on local measurements at the extruder exit by an optical sensor. The information extracted from the thermomechanical characterisation of the bulk polymer (i.e., DSC, TGA, theology) was used as a good first order approximation to establish the extruding conditions. The final settings of the filament maker, shown in Table 1, were ultimately established through trial-and-error. The methodology used was further validated with a different polymer (not shown in this work).

### 2.4. 3D Printing

The 3D printer used was the Ultimaker 2+ (Ultimaker, Utrecht, The Netherlands). The print-head was replaced by an E3D Titan extruder with a V6 hot-end and a 0.8 mm nozzle. In order to help the material to maintain stability after exiting the nozzle, an additional fan was installed and a duct was attached to direct the air towards the nozzle exit. G-code files for printing were generated using CURA software (4.3.0) (Ultimaker, Utrecht, The Netherlands) to print rectangular blocks measuring 20 mm × 10 mm × 4 mm.

Ninjaflex samples were printed under three different printing conditions. The printing speed was kept constant at 20 mm/s, the layer height was set at 0.4 mm, and the print-bed temperature at 30 °C. One sample was printed at 225 °C print-head temperature and 0.8 mm infill distance. The second sample was printed at 235 °C print head temperature and 0.8 mm infill distance. The third sample was printed at 235 °C and at an infill distance of 0.5 mm. 

SH-TPU samples were printed at a print speed of 20 mm/s at three different print-head temperatures: 225 °C, 230 °C, and 235 °C. The print bed temperature was kept at 30 °C, the infill distance was constant at 0.8 mm and the fan was operated on full capacity. Before printing, the diameter of the filament was measured over the length required to print the part (40 cm). Filaments were only used if the average diameter was between 1.5 and 1.8, and the standard deviation in the diameter was below 0.1 mm. The extrusion rate was automatically compensated for the thickness of the filament. At each condition, two samples were printed. 

### 2.5. Mechanical Testing and Healing Test

The mechanical properties of the 3D printed samples were evaluated using a compression-cut test based on a previously reported test for elastomers [20]. This test was implemented since it allows for obtaining (quasi)quantitative information on polymers available in small quantities. The mechanical tests were performed using an Instron Model 3365 universal testing system (Instron, Norwood, MA, USA) with a 1kN load cell. Before the test, the 3D printed parts and bulk polymer were cut into 4 × 4 × 4 mm cubes. During the test, a blade with the following dimensions was used: the angle of the tip was 18°, the length of the tip was 0.75 mm, and the width of the rest of the blade was 0.20 mm. The razor blade was then brought in contact with the sample and allowed for displacing vertically against the sample at a constant speed of 10 mm/s. The test stopped automatically at 3.8 mm displacement to prevent any damage. All tests were performed at 20 °C. The test set-up can be seen in Figure 2a. The samples tested during the compression-cut test were recorded using a Celestron handheld digital microscope (Celestron, Torrance, CA, USA). 

3D printed samples were tested in two directions, the *xz-* and the *xy*-direction. These were selected as they typically give the most extreme results in terms of anisotropy [5,6,7]. Figure 2b shows a schematic of the samples with the axis system used during the test. The layers are stacked in the *z* direction during printing. During the mechanical test in the *xz*-direction, the sample was positioned such that the blade moved along the *z* direction in the *xz*-plane (Figure 2c). During the test in the *xy*-direction, the blade was positioned to move along the *x*-axis parallel to the *xy*-plane (Figure 2d). Two samples were tested in each direction. 

The self-healing behaviour of the TPU and the SH-TPU samples was tested using the same mechanical testing protocol described above. This allowed for creating the damage (cut) in a controlled manner and testing the mechanical behavior of the healed samples in a comparable manner. The healing protocol consisted of leaving the cut samples without any external pressure in an oven at 30 °C for 24 h. The healing time and temperature were based on the results reported in a previous work [25]. In [25], the ability of the SH-TPU, named MDI-p in this previous work, to fully heal cut-through damages and recover fracture resistance properties was fully demonstrated. At the selected healing temperature, interfacial healing rapidly occurs by hydrogen bonding swap among urethane units while full mechanical recovery is achieved at longer times due to chain interdiffusion and establishment of supramolecular interactions at the interphase.

## 3. Results and Discussion

### 3.1. Material Characterisation

Figure 3 shows the FTIR spectra of SH-TPU as produced (neat polymer), as filament and as a 3D printed sample. In agreement with previous works [25], the broad peak at 3325 cm^−1^ is associated with the stretching vibration of hydrogen bonded N-H groups, or Amide II band, the sharp peak at 2925 cm^−1^ corresponds to the stretching of C–H bonds in aliphatic compounds and the amide I band at 1700 cm^−1^ reflects the stretching vibrations of C=O bonds. The absence of a peak at 2250 cm^−1^ confirms the absence of unreacted isocyanate groups. No significant changes in the spectra were observed after the polymer processing into the filament and 3D printed part.

TGA was used to establish the upper limit temperature of the SH-TPU during printing. Figure 4a shows the results of the TGA measurements of the bulk and filament SH-TPU. From the weight loss curve, a thermal degradation temperature at 0.2% mass loss (T_d0.2%_) of 290 °C was deduced and attributed to the thermolysis of the urethane bonds as indicated in previous works for other PUs degrading at similar temperatures [26]. Since the mass loss deviated from 0 at 250 °C, this temperature was chosen as the maximum processing temperature. The TGA curve for the filament slightly shifted to higher temperatures with a slight drop of the degradation temperature at 0.2% mass loss (~260 °C). 

Figure 4b shows the DSC results of the first heating of SH-TPU, in bulk and filament form. For the bulk polymer, the glass transition temperature (T_g_ = 5.5 °C) was determined using the inflection point method, and coincided with previously reported T_g_ values [25]. Additionally, a small endothermic event, not reflected in the cooling curve, was observed in the first heating curve at around 205 °C (indicated with an arrow in Figure 4b). As this peak was reproducible across measurements and even in a different DSC device, it cannot be attributed to an artefact. Based on previous works on segmented thermoplastic polyurethanes, we attribute this endothermic peak to the dissociation of hydrogen bonds and aromatic interactions in the hard blocks [27]. The DSC of the filament showed an endothermic peak at ~50 °C, attributed to partial crystallinity resulting from the rapid cooling during filament making.

Once the upper temperature was fixed, rheology was performed to select the most appropriate temperature range for printing. Figure 5 shows the results of the temperature sweep rheology. The elastic modulus (G’) and the viscous modulus (G”) crossed at roughly 195 °C. This temperature was close to the endothermic event observed during DSC measurements at 205 °C, which was associated with dissociation of reversible groups in the hard block. Above this crossover temperature, the polymer behaves like a viscous liquid, thereby establishing the lower temperature limit for melt-processing into a filament and during the 3D printing.

Figure 6 compares the viscosity (η) of Ninjaflex (a) and the SH-TPU used in this work (b) as a function of the shear rate ($\overset{\cdot }{ɣ}$) for different temperatures above the crossover point temperature shown in Figure 5. For all the temperatures used, the viscosity of Ninjaflex polymer remained constant with the shear rate until a shear rate between 10 and 100 s^−1^, depending on the temperature, was reached. At this point, the viscosity suddenly dropped and became highly temperature independent. As can be seen in Figure 6b, the SH-TPU did not display this sharp drop, and maintained a relatively high viscosity at high shear rates at all the studied temperatures.

The printing temperature range needs to be set considering a good balance between viscosity and shear rate. To establish these parameters for the SH-TPU, we used the printing temperature range for Ninjaflex as guidance. Experimentally, Ninjaflex could be printed already at temperatures above 205 °C. Nevertheless, the quality of the printed parts was acceptable when the temperature was above 225 °C and dropped when the temperature was above 235 °C. Comparing the viscosity of Ninafjex and the SH-TPU at low shear rates (below 100 s^−1^) revealed that 225 °C was likely the lowest printing temperature possible for SH-TPU since; at that point, the viscosity drops under 1000 Pa·s. Interestingly, at 235 °C, the viscosity of the SH-TPU is largely shear-rate independent and remains relatively high, compared to Ninjaflex. This feature can be used for 3D printing as a feature to allow printing without too high viscous flow at the printed part while fast cooling will rapidly increase the viscosity, thereby helping to maintain the shape integrity of the printed part. At 240 °C, rheology tests failed at high shear rates due to loss of polymer integrity, thereby establishing the melt processing, and printing, range for the SH-TPU also between 225 and 235 °C. 

### 3.2. Filament Making

The SH-TPU filaments obtained with the filament maker can be observed in Figure 7a. Unfortunately, the limited versatility of the filament maker limits the ultimate quality of the filaments obtained despite the fine-tuning attempts. In particular, the small distance between the entrance and the first heated section of the extruder screw resulted in agglomeration of the pellets at the entrance and hence in an irregular inflow of material and air entrapment. As a result, the filament showed thickness variations along the length. The variation of the diameter throughout the length of the filaments used to print the samples can be seen in Figure 7b for each filament used in each condition during the 3D printing process. The diameter of the Ninjaflex filament was consistently 1.70 mm. It is expected that further improvements in the filament making process (optimization) will lead to higher filament quality, but this was out of the scope in this work.

### 3.3. 3D Printing SH-TPU

Figure 8 shows pictures of 3D printed SH-TPU samples. A significant difference in quality can be observed between the samples. The samples printed at 225 °C were of relatively poor quality, with a high porosity, visible on the outside of the specimen, and in agreement with the higher viscosity during printing as given by Figure 6b. Samples printed at 230 °C and 235 °C looked more regular, where the samples printed at 230 °C were of the best quality based on visual inspection. The samples printed at 235 °C showed some over-extrusion at the right end. The printing of the SH-TPU led to comparable chemical composition to the bulk polymer as shown in Figure 3. Additionally, a more complex hollow vase structure could be 3D printed using SH-TPU (shown in the Appendix A).

#### Microscopy Imaging

Figure 9 shows microscopy images of the 3D printed samples. In the Ninjaflex samples (Figure 9a), the void locations follow a clear pattern and show a high directionality. There was a clearly visible boundary between each of the deposited filaments. For the samples printed with a 0.8 mm infill distance, the stacked rows of filament did not always touch the next row. This can be seen especially well as holes in the pictures of the *xz*-plane and as the presence of alternating matte and glossy regions in the *yz*-plane, where the matte areas can be attributed to cuts in the material and the glossy regions to areas where there was no contact between filaments. This contact improved with the higher printing temperature and with a smaller infill distance (samples at 235 °C/0.5 infill). The void size also decreased with printing temperature and infill distance, thereby reflecting an overall improvement of the printed parts; nevertheless, at temperatures above this range, the samples lost integrity due to excessive flow.

The SH-TPU samples (Figure 9b) showed a significantly lower void content already at low temperatures with no clear boundaries between the individual deposited filaments visible even if the infill was set at 0.8 mm. In addition, the voids are random in shape and location with a slight directionality in the *y*-direction. Due to their random distribution and general high homogeneity of the samples, the voids cannot be attributed to an insufficient fusion between filaments as occurred in the Ninjaflex 3D printed parts. Instead, these defects are here attributed to the improvable quality of the filaments used during printing as they showed uneven thickness and the presence of entrapped air as discussed in the filament making section. The directionality, size, symmetry, and abundance of the voids reduced with the printing temperature reaching the almost complete disappearance of defects at 235 °C in samples which still maintained a high shape integrity. Despite the presence of defects due to the filament quality, this result confirms the potential of self-healing polymers to obtain high quality 3D printed parts with no obvious detection of the filaments that built the sample. 

### 3.4. Mechanical Testing

Figure 10 shows the typical force-displacement response of an SH-TPU sample and a 3D printed Ninjaflex sample during the compression cut test with arbitrary units. The mechanical response of each polymer is accompanied with micrographs corresponding to each displacement in an attempt to facilitate the interpretation of the force–displacement curves obtained in the used testing conditions. 

The curve for SH-TPU can be divided into four regions after analysing the force–displacement curve in combination with the micrographs. The first region (a–b) is a linear compression region, where the base area of the sample increased linearly and most deformation occurred at the top region of the sample in contact with the blade. The second region (b–c) is a nonlinear deformation due to the rapid deformation of the sample at its base and in the entire sample itself. In this region, the force–displacement response depends not only on the bulk properties of the sample, but also on the interlayer behaviour within the polymer subject to shear and tension. When testing in the *xz*-direction, these forces occur between the stacked layers of the print and therefore the response also depends on the interlayer adhesion. In the *xy*-direction, these forces occur between different rows in the print. The third region (c–d) shows again linear compression as the base area expansion rate was lower than in b–c. In the last region (d–e), the force reaches its maximum and unveils the sample ultimate failure. 

In the Ninjaflex sample, the same regions can be identified: an initial linear region (f–g), followed by a less-obvious nonlinear region (g–h), a second linear region (h–i) and finally a maximum force indicating failure (i–j). The nonlinear transition region (g–h) of the Ninjaflex sample was less pronounced than in the SH-TPU, and the shape of the sample also showed less deformation in agreement with the stiffer response of Ninjaflex compared to SH-TPU. The results were reproducible between samples produced and tested at the same conditions (see Appendix A).

The analysis of the compression cut test allowed the interpretation of the results obtained for both the Ninjaflex and the SH-TPU polymers printed at different conditions. In general terms, the first region (a–b or f–g) shows the response of the sample as a whole, the second region (b–c or g–h) is affected by the geometry and interlayer and interfilament properties, and the last region shows the force at break during compression-cut test (d–e or i–j).

Figure 11 and Figure 12 show representative results for Ninjaflex and SH-TPU, respectively. The whole set of results can be found in the Appendix A. Table 2 summarizes some key values that could be extracted from the load–displacement curves as an average between the two tests for each condition. 

Figure 11 shows the results of 3D printed Ninjaflex. Initially, up to 1.5 mm displacement, the responses were highly similar for all printing conditions and cut plane direction. An exception was the *xz*-direction of the sample printed at 235 °C and 0.5 mm which showed a higher slope in this region. At 1.5 mm, the samples printed at 235 °C and 0.8 mm showed a constant 40 N force as the cut progressed through the sample. The other samples failed at near two times higher loads (80–90 N) and displacement (3.2 mm). The sample printed at 235 °C and 0.5 mm and tested in the *xz*-plane deviated from the rest, since it failed at an even higher load (115 N), but at a similar displacement (3.2 mm) suggesting a higher quality of the print. 

The load–displacement results are in good agreement with the quality of the prints shown in the micrographs in Figure 9a. The samples with the largest void area (235 °C, 0.8 mm) also resulted in the poorest mechanical response independent on the cut plane. The sample printed at 225 °C and 0.8 mm showed a higher mechanical response in agreement with a lower void content, yet still independent on the cut plane. The sample printed at 235 °C and 0.5 mm shows comparable, yet smaller, voids in the *xy*-plane but significantly lower defects in the *xz*-plane. The higher maximum load for this sample in the *xz*-direction can be therefore explained by the good fusion between filament planes perpendicular to the cut (Figure 9a) while the lower properties in the *xy*-direction, comparable to the sample 225 °C/0.8mm, can be attributed to the limited interfilament fusion parallel to the cut, which was also seen in Figure 9a.

The results suggest that in samples showing a poor cross-section quality with a high void concentration, the mechanical response is dominated by the high void content and irregularity. The sample printed at 235 °C/0.5 mm infill distance showed a more homogenous geometry in the *xy*-plane, but the presence of voids between filament rows resulted in a significant difference in load–displacement behaviour between the two cut directions. The similar results in the *xy*-direction independent of the printing conditions and the significantly improved response in the *xz*-direction for the 235 °C/0.5 mm sample suggest an improvement in the interlayer adhesion without significant improvement in the interfilament row adhesion in the *xy*-plane, thereby leading to an increased anisotropy. The isotropy found in the other two printing conditions can be attributed to the imperfection of the print and dominance of the defects on the mechanical behaviour rather than to an improved printing. 

Figure 12 shows the mechanical response of the SH-TPU samples to the compression cut test. Overall, the mechanical response of the SH-TPU samples was comparable to that of the bulk SH-TPU material shown as an example in Figure 11 and in the reference black line in Figure 12. Nevertheless, the second region of the plot, the nonlinear compression, was more extended in the displacement for the 3D printed samples. Failure of SH-TPU was not captured in the current testing conditions since the test ended before the full cut. Table 2 therefore only shows the failure load and failure displacement of the bulk SH-TPU samples. Considering that the sample dimensions were the same in all cases, the failure load of the print samples is expected to be similar or, given the force–displacement curves, just slightly higher to that of the bulk material.

Up to 1.5 mm displacement, the responses of the samples were highly similar independently of the printing conditions and direction of the cut. A clear exception is for the *xz*-direction of the sample printed at 235 °C/0.8 infill, which shows a significantly stiffer behaviour. For the sample printed at 225 °C, there was no significant difference between the *xz*- and the *xy*-response. This isotropy can be attributed, as for the Ninjaflex, to the governance of the defects visible in Figure 9b. For the other two printing conditions, the difference was more pronounced. Notably, the sample printed at 230 °C showed a lower mechanical response for the *xy*-direction, whereas the *xz*-response was still similar to 225 °C. For the sample printed at 235 °C, the *xz*-response became higher. The higher force-displacement values of the 235 °C are in agreement with the micrographs shown in Figure 9b, with this sample being the most homogeneous of all and suggesting an improved interlayer adhesion perpendicular to the *xz*-cut plane at this temperature. The values of the *xy*-plane being similar to those of the other printing conditions cannot be directly explained from the micrographs in Figure 9b as these suggest homogeneous samples in both cut planes with high interfilament fusion and lack of defects besides randomly distributed holes. The justification for the anisotropy in this case might be found at the molecular level. While the SH-TPU clearly leads to a better filament fusion and interlayer adhesion, the mechanical results suggest that the interaction forces induced by intermolecular diffusion and hydrogen bonds and aromatic interactions between filaments parallel to the *xy*-plane is lower at 230/235 °C than those obtained at lower printing temperatures (225 °C shows higher force–displacement). This result may indicate a reversed bonding effect at increased temperatures that falls outside the scope of this work yet worth exploring in future works with self-healing polymers under 3D printing. 

#### Self-Healing Behaviour

Figure 13 shows the results of the healing experiments for the *xz*-cut-plane by comparing the load–displacement of pristine and damaged-and-healed samples. The healing tests were performed on 3D printed samples showing the lowest anisotropy that still has good mechanical properties. It turned out that, for both polymers, these requirements are met in samples print at 225 °C and 0.8 mm infill distance, as seen in Figure 11 and Figure 12. In Figure 13a,b, the vertical dotted black lines indicate the depth of the cut that was performed to the samples before healing in relation to the displacement during the test. 

In Figure 13a, it can be clearly seen that the Ninjaflex was unable to restore the damage, since the force of the healed sample is well below that of the pristine sample. The measured force during the initial displacement of the “healed” sample (below 2 mm) is attributed to the friction of the crack walls with the blade during displacement. When the blade reached the bottom of the crack, at roughly 2.3 mm, the force increased as a new compression cut test was being initiated in the pristine region of the “healed” polymer.

Figure 13b shows the high levels of damage restoration achieved in both the as-produced bulk SH-TPU and the 3D printed SH-TPU. This indicates that the 3D printed SH-TPU retained its healing ability even after processing and printing, with the compression cut yielding identical results after healing for both the bulk and the 3D printed sample. Although the overall force–displacement curve of the 3D printed part with the SH-TPU is lower than that of the commercial polymer, the restoration of the mechanical properties after damage offers a clear benefit for life extension during use together with increased isotropy due to better interfilament fusion at both plane and row level. The full results and micrographs of the healing experiments can be found in the Appendix A. 

## 4. Conclusions

In this work, a protocol for the FDM printing and testing of self-healing polymers is introduced using a previously reported self-healing thermoplastic polyurethane. The self-healing polyurethane (SH-TPU) was synthesised into a slab and transformed into a continuous filament to be used in filament deposition modelling techniques. The large variations in the SH-TPU filament diameter resulted in an uneven polymer deposition during printing which reduced the quality of the prints below its full potential and should be matter of further optimization. Nevertheless, for the same infill distance, at certain printing temperatures, the 3D printed SH-TPU showed very low void content and the absence of the filament marks in all plane directions, compared to a commercial 3D printing polyurethane, still maintaining full shape integrity of the print part. 

A compression cut test and analysis protocol was introduced for the (quasi)quantitative evaluation of the mechanical properties and healing behaviour of polymers produced in small volumes. The load–displacement results of the compression-cut test could be better interpreted with the help of in-situ micrographs and correlated to the quality of the prints. The 3D printed SH-TPU exhibited a mechanical behaviour that did not depend significantly on the printing conditions or direction of the cut-plane. Moreover, the SH-TPU completely retained its self-healing ability after the 3D printing process, showing full restoration of the mechanical behaviour. Although the mechanical properties of the SH-TPU were slightly below those of the commercial one, the results demonstrate the potential of healing polymers to obtain improved printing quality with increased interfilament fusion (between planes and rows) and with the added value of repairing macroscopic damages affecting full sample integrity. 

## Figures and Tables

**Figure 1 polymers-13-00305-f001:**
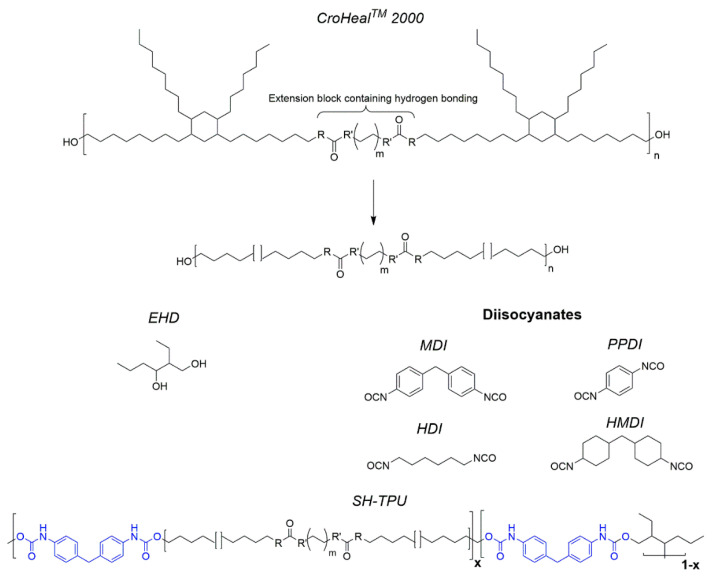
Chemical structure of CroHeal™ 2000, EHD and MDI and the molecular structure of the segmented SH-TPU used in this work with the soft block containing hydrogen bonds and aliphatic side branches, and the hard blocks containing aromatic interactions and hydrogen bonds.

**Figure 2 polymers-13-00305-f002:**
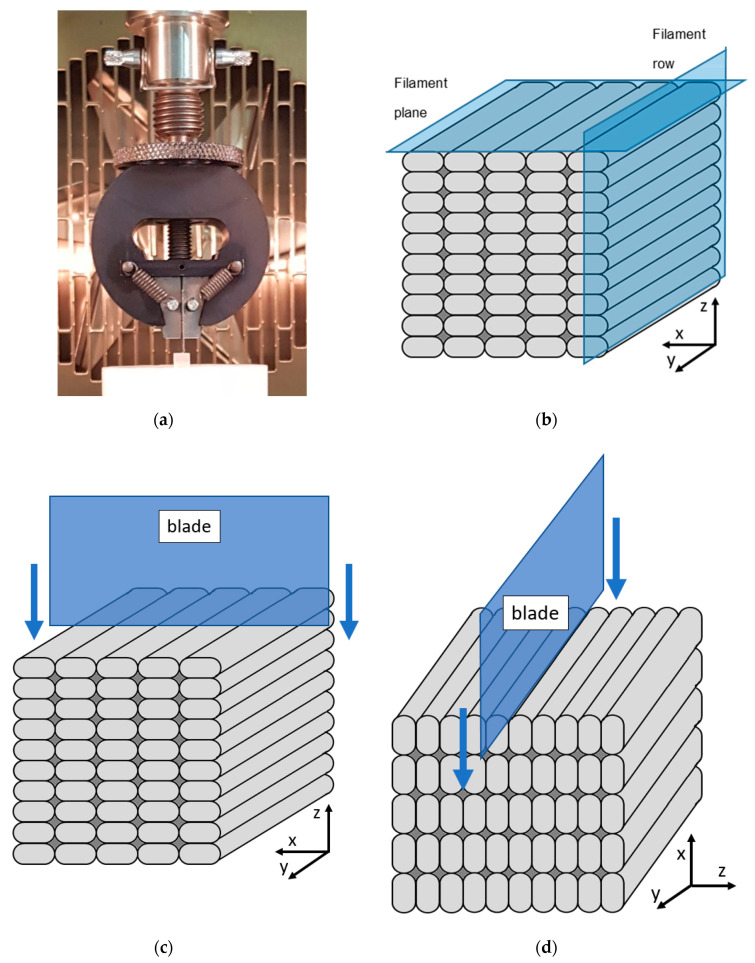
(**a**) displacement clamp with blade and sample (white cube) as used during the mechanical testing; (**b**) schematic of the 3D printed sample; (**c**) schematic of the compression cut test in the *xz*-direction; (**d**) schematic of the compression cut test in the *xy*-direction.

**Figure 3 polymers-13-00305-f003:**
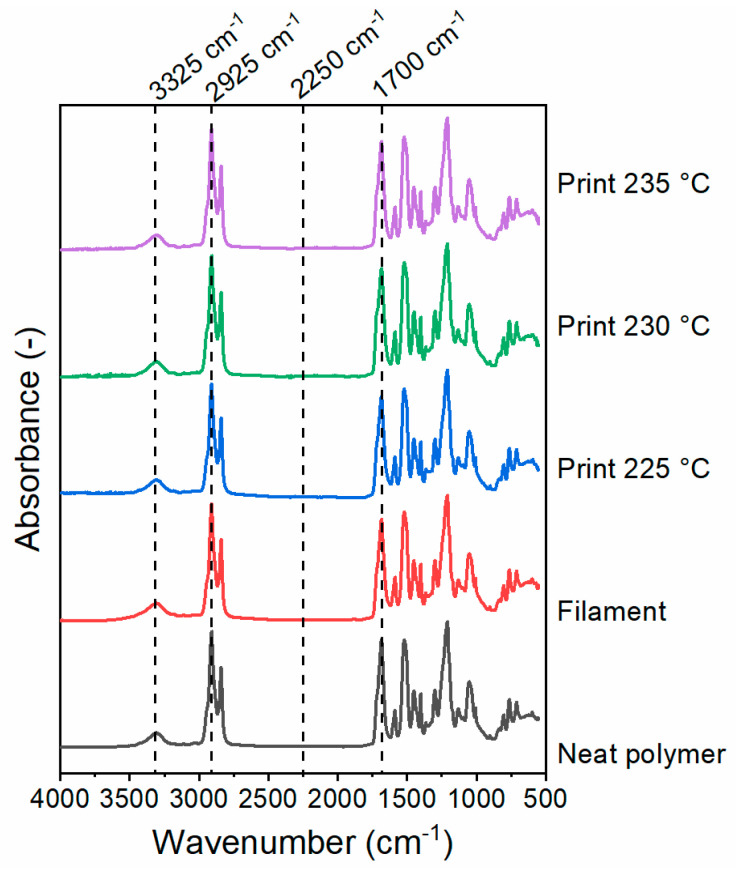
FTIR spectra of the SH-TPU as produced (neat polymer), as filament and as 3D printed polymer at different printing temperatures (225, 230 and 235 °C).

**Figure 4 polymers-13-00305-f004:**
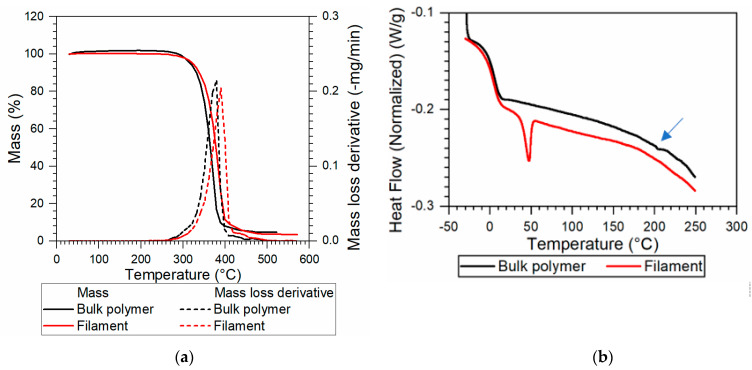
Thermal analysis results of the SH-TPU in the bulk and the filament form. (**a**) TGA at 5 °C/min; (**b**) DSC at 5 °C/min (endo down); the blue arrow indicates the reproducible endothermic peak at near 200 °C.

**Figure 5 polymers-13-00305-f005:**
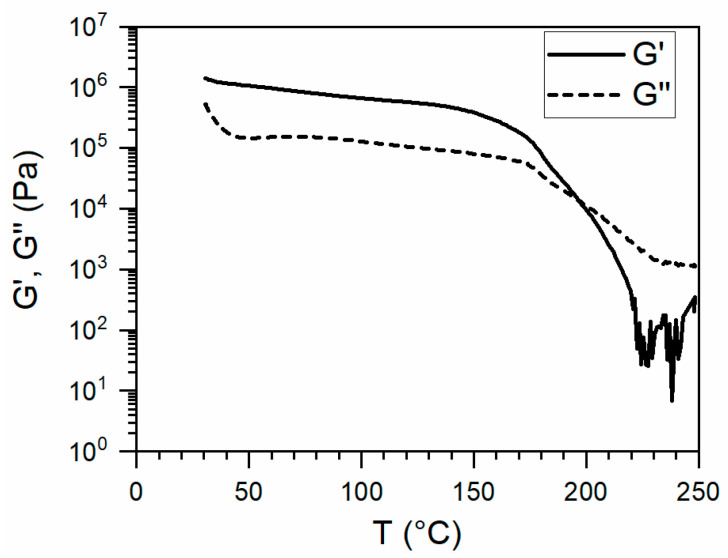
Temperature sweep rheology results of bulk SH-TPU at 1 °C/min and 1 Hz.

**Figure 6 polymers-13-00305-f006:**
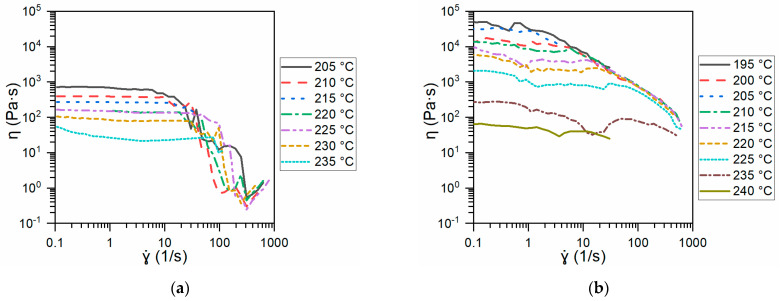
Viscosity (η) as function of the temperature and the shear rate ($\overset{\cdot }{ɣ}$) for (**a**) Ninjaflex and (**b**) the SH-TPU.

**Figure 7 polymers-13-00305-f007:**
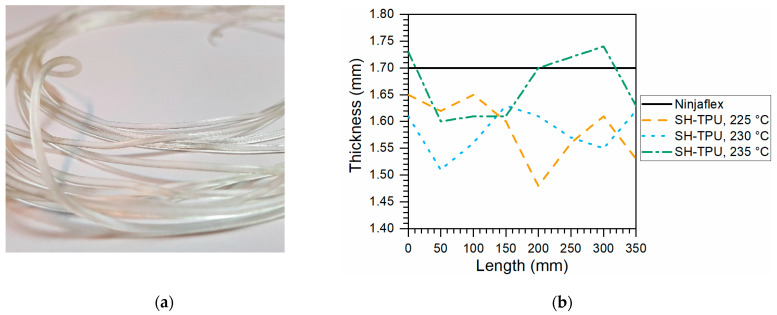
SH-TPU filament making results. (**a**) picture of the extruded SH-TPU filament; (**b**) thickness distribution of the filaments used for printing.

**Figure 8 polymers-13-00305-f008:**
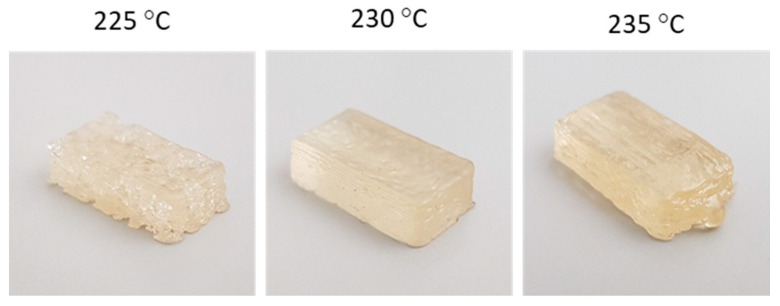
Pictures of the 3D printed SH-TPU samples of approximately 10 × 20 × 5 mm.

**Figure 9 polymers-13-00305-f009:**
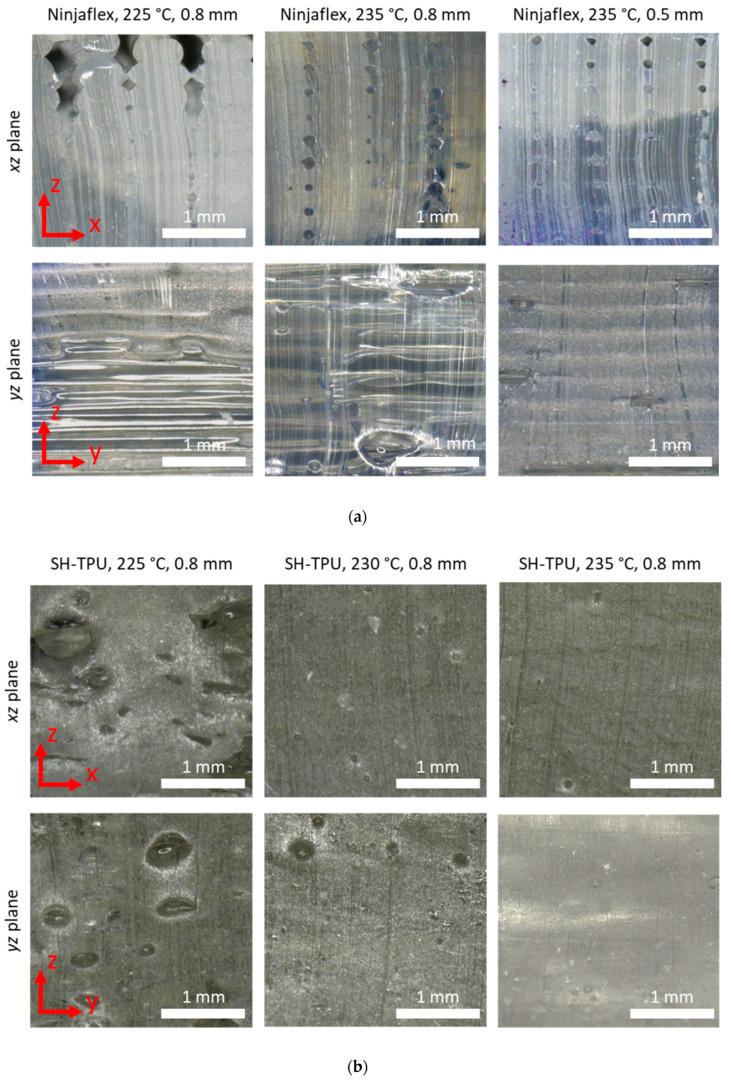
Microscopy images of the 3D-printed samples at different printing conditions: (**a**) Ninjaflex and (**b**) SH-TPU. The *xz*-plane shows defects in between filament planes, and the *yz*-plane shows defects between filament rows.

**Figure 10 polymers-13-00305-f010:**
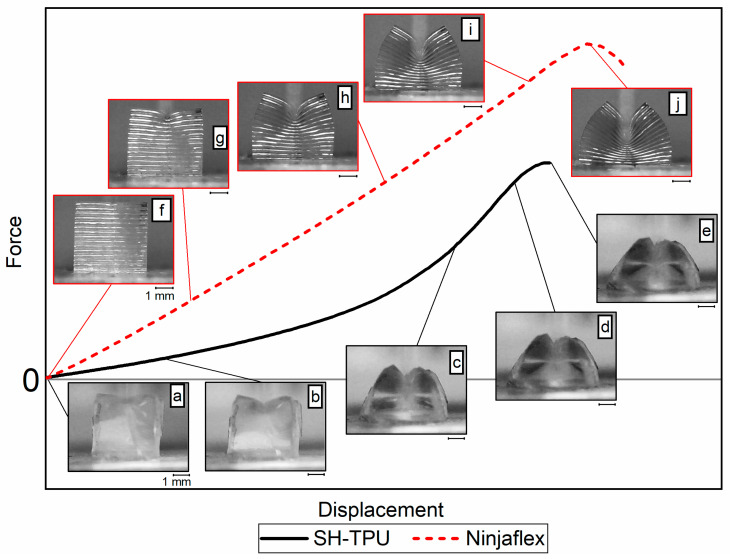
Typical mechanical response during the compression cut test including micrographs describing each load-displacement region.

**Figure 11 polymers-13-00305-f011:**
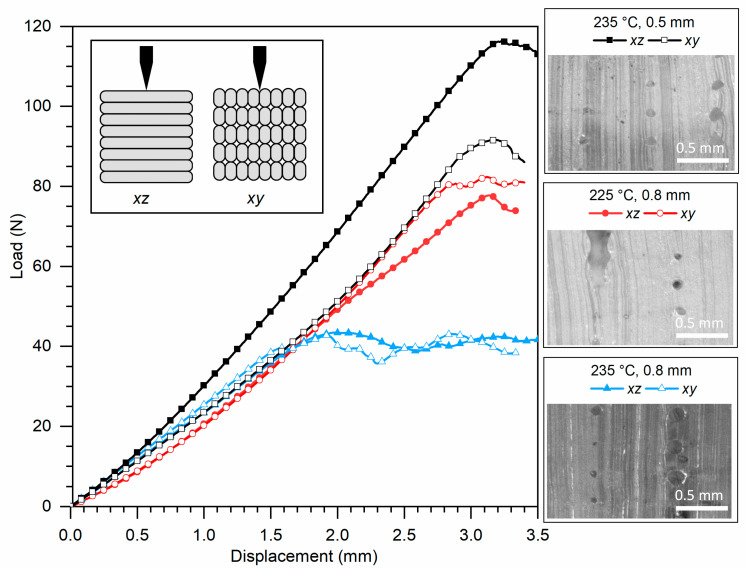
Compression cut test results for 3D printed Ninjaflex under three print conditions: 225 °C with a 0.8 mm infill distance, 235 °C with a 0.8 mm infill distance and 235 °C with a 0.5 mm infill distance. The graph is accompanied by microscopy images of the cross-section in the *xz*-plane. Scale bars represent 0.5 mm.

**Figure 12 polymers-13-00305-f012:**
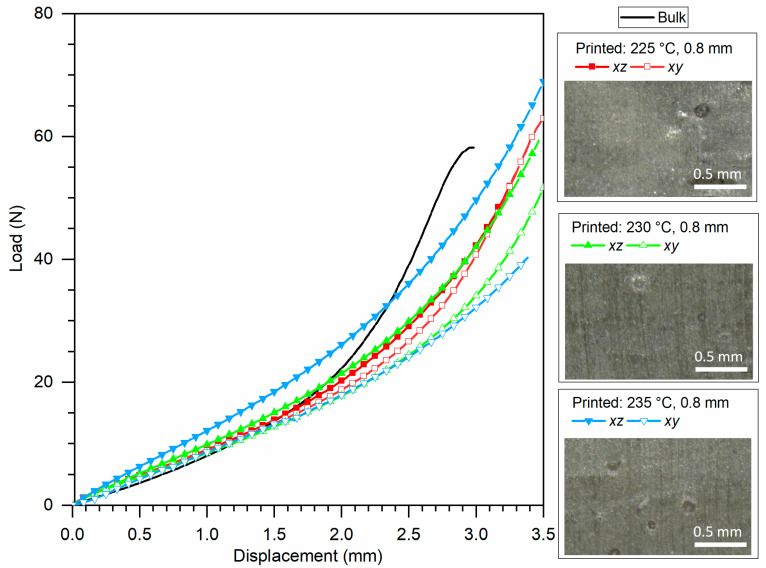
Comparison of the mechanical testing of bulk SH-TPU to 3D printed SH-TPU at 225 °C, 230 °C, and 235 °C with constant infill distance of 0.8 mm. The graph is accompanied by microscopy images of the cross-section in the *xz*-plane. Scale bars represent 0.5 mm.

**Figure 13 polymers-13-00305-f013:**
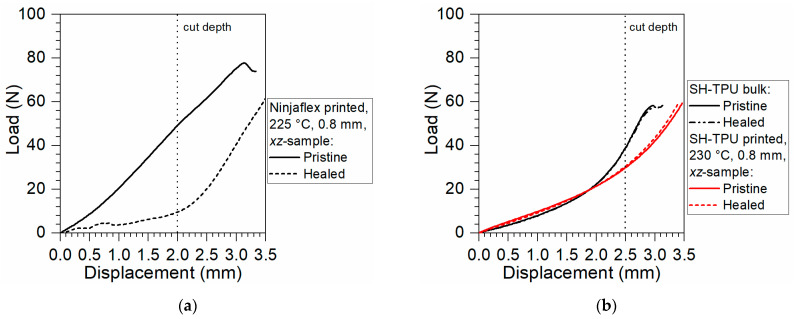
Compression-cut mechanical characterisation of the healed (**a**) Ninjaflex and (**b**) SH-TPU polymers healed at 30 °C/24 h and tested at 20 °C, 10 mm/s. Vertical dotted lines indicate the position of the cut created during the first compression-cut test that was subsequently healed.

**Table 1 polymers-13-00305-t001:** Settings of the 3DEVO Next 1.0—ADVANCED filament maker. T1, T2, T3, and T4 represent the temperatures of the different heating zones of the extruder section of the filament maker, where T1 is the extruder section closer to the exit.

Extruder Entrance T4	T3	T2	Extruder Exit T1	Screw Rotation Speed	Percentage Fan Power	Target Filament Thickness
200 °C	215 °C	220 °C	210 °C	6.5 rpm	60%	1.7 mm

**Table 2 polymers-13-00305-t002:** Analysis of the force displacement curves in Figure 11 and Figure 12.

Sample		Slope Linear Part		Failure Load		Failure Displacement	
		Slope (N/mm)	*xz*/*xy* (%)	Load (N)	*xz*/*xy* (%)	Displacement (mm)	*xz*/*xy* (%)
Ninjaflex, 225 °C, 0.8 mm	*xz*	21.3	99.8	77.8	101.6%	3.13	106.2%
	*xy*	21.4		76.5		2.95	
Ninjaflex, 235 °C, 0.8 mm	*xz*	25.8	104.1	45.1	122%	1.89	116.5%
	*xy*	24.8		69.9		1.60	
Ninjaflex, 235 °C, 0.5 mm	*xz*	30.9	134.4	114.7	125.1%	3.22	101.0%
	*xy*	23.0		91.7		3.16	
SH-TPU bulk	-	7.5	N.A.	60	N.A.	2.90	N.A.
SH-TPU, 225 °C, 0.8 mm	*xz*	8.0	111.7	>60	-	>3.50	-
	*xy*	7.1		>60		>3.50	
SH-TPU, 230 °C, 0.8 mm	*xz*	9.7	125.4	>60	-	>3.50	-
	*xy*	7.7		>50		>3.50	
SH-TPU, 235 °C, 0.8 mm	*xz*	12.1	135.1	>70	-	>3.50	-
	*xy*	8.9		>70		>3.50	

## Data Availability

The data presented in this study are openly available in 4TU.ResearchData at doi.org/10.4121/13603775.

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
