# Peer review of "3D Printing of a Self-Healing Thermoplastic Polyurethane through FDM: From Polymer Slab to Mechanical Assessment"

_polymers, 2021, doi:10.3390/polym13020305_

Round 1
Reviewer 1 Report
This is a well structured and well written paper, that can open the way for interesting applications. It surely deserves publication, i only recooment the following minor revision:
1) Line 213. Put the reference in the final form.
2) Line 341. Perhaps you should substitute "Figure 10a" with "Figure 10"
Reviewer 2 Report
In this paper, the authors present the complete process for fused deposition modelling of a room temperature self-healing polyurethane including synthesis, 3D prining, characterization and assessment.
This paper has clear logic and detailed experiments which can support the conclusions with reference value for peers and the field. I recommend it accepted in present form.
Author Response
Please see the attachement.

Reviewer 3 Report
The introduction can be improved providing more references on self-healing PU. This will give to the readers a better overview on mechanical and SH performances of similar systems.
Ref 18 and 22 are the same, please correct.
Reviewer 4 Report
General comments:
This manuscript delineates a novel methodology for creating 3D printable self-sealing materials. The manuscript is well-written, however the reviewer thinks that the manuscript can be further improved if the following issues are addressed.
Specific comments:
1. Table 1. The authors show 2 process parameters, namely screw rotation speed and fan speed. The reviewer thinks that the author should also discuss how these process parameters relates to the physical process in a more scientific way, such as the extrusion speed and heat loss.
2. Missing scale bars in figures 8,11, and 12.
3. Table 1. The way temperature parameter is presented looks odd. the temperatures should be grouped into a single column and displayed in separate rows for different temperatures.
4. is there any test standard for cut test? or is it adopted from any standard? can the authors explain how the test method came about? are there any references?
5. did the authors do the cut test for YZ plane? if yes, why did the authors not include the result? if no, why the test is not conducted?
6. figure 4 shows the thermal analysis for bulk SH-TPU. Did the author do the thermal analysis for extruded SH-TPU (that were extruded at different temperatures)? If no, why not?
7. can the author emphasize the healing mechanism and the conditions required for the healing to happen?
8. How long does the healing process take? it is mentioned 30 degree Celcius and 24 hours? how did the authors arrive at these values? any justifications?
